# A Two-Layer Self-Organizing Map with Vector Symbolic Architecture for Spatiotemporal Sequence Learning and Prediction

**DOI:** 10.3390/biomimetics9030175

**Published:** 2024-03-13

**Authors:** Thimal Kempitiya, Damminda Alahakoon, Evgeny Osipov, Sachin Kahawala, Daswin De Silva

**Affiliations:** 1Centre for Data Analytics and Cognition, La Trobe University, Melbourne, VIC 3086, Australia; s.kahawala@latrobe.edu.au (S.K.); d.desilva@latrobe.edu.au (D.D.S.); 2Department of Computer Science, Electrical and Space Engineering, Luleå University of Technology, 971 87 Luleå, Sweden; evgeny.osipov@ltu.se

**Keywords:** self-organizing maps, spatiotemporal sequence learning, vector symbolic architectures, hierarchical temporal memory

## Abstract

We propose a new nature- and neuro-science-inspired algorithm for spatiotemporal learning and prediction based on sequential recall and vector symbolic architecture. A key novelty is the learning of spatial and temporal patterns as decoupled concepts where the temporal pattern sequences are constructed using the learned spatial patterns as an alphabet of elements. The decoupling, motivated by cognitive neuroscience research, provides the flexibility for fast and adaptive learning with dynamic changes to data and concept drift and as such is better suited for real-time learning and prediction. The algorithm further addresses several key computational requirements for predicting the next occurrences based on real-life spatiotemporal data, which have been found to be challenging with current state-of-the-art algorithms. Firstly, spatial and temporal patterns are detected using unsupervised learning from unlabeled data streams in changing environments; secondly, vector symbolic architecture (VSA) is used to manage variable-length sequences; and thirdly, hyper dimensional (HD) computing-based associative memory is used to facilitate the continuous prediction of the next occurrences in sequential patterns. The algorithm has been empirically evaluated using two benchmark and three time-series datasets to demonstrate its advantages compared to the state-of-the-art in spatiotemporal unsupervised sequence learning where the proposed ST-SOM algorithm is able to achieve 45% error reduction compared to HTM algorithm.

## 1. Introduction

The digital age and the wide use of technology have generated massive volumes of information in digital form, which in most situations is present in the form of data streams that contain spatiotemporal patterns in sequence form depicting real-life events [1]. The wide availability of spatiotemporal sequence information in various forms generated via diverse applications such as videos, social media, news streams, Industry 4.0 processes, and smart environments has vastly increased the importance and applicability of algorithms that can learn and predict based on spatiotemporal sequences [2]. Many artificial intelligence (AI) and machine learning algorithms have been developed to capture, learn, and build models using spatiotemporal information from this data, and a few algorithms have even focused on predicting the next events. The high volatility of data generated and the tendency for frequent change in behaviors in these environments have made it difficult to rely purely on the stability of spatiotemporal patterns from past data to inform future behaviors. The concept of predictive learning, therefore, becomes important where a machine learning algorithm needs to have the ability to augment and adapt its past learning based on outcomes of continuous prediction of future events [3]. Therefore, the fast-changing and dynamic nature of these environments represented by volatile and frequently changing spatiotemporal patterns has imposed several key requirements from machine learning algorithms:(a)Unsupervised learning ability for learning without pre-labeled data.(b)Representational learning of spatial and temporal elements as separate building blocks for the construction of spatiotemporal patterns and adapt to changing environments.(c)Ability to represent and manipulate temporal sequences of variable length (order).(d)Continuous online prediction of the next occurrence of a sequence based on sequential recall of a pattern from memory.

Although a number of research innovations including new algorithms have been targeted at addressing these requirements, none of these have successfully addressed all the requirements, especially the ability of real-time prediction.

In this paper, a novel algorithm is proposed to address the above listed requirement. The new algorithm is built upon a foundation of two layers of self-organizing maps (SOMs), which provides the ability of unsupervised learning as well as a topology preserving mapping of the input data space [4,5]. The two SOM layers cater to separate representational learning of spatial and temporal elements based on the spatiotemporal patterns in the input space. The spatial elements captured in the first of the two layers act as an alphabet with which the temporal sequences are constructed and represented in the second layer. The separation of the layers allows the representation of the spatial alphabet independent from a particular pattern or sequence and therefore can be used to represent the spatial elements of other sequences that contain the same spatial elements in a different order. The requirement of the fixed dimension in the input data in traditional SOMs is a limitation when representing real-life sequences of variable length, and the proposed algorithm resorts to vector symbolic architectures (VSAs) to address this limitation. A memory module facilitates the continuous prediction of the next elements of sequences based upon spatial elements captured in the first SOM layer. The topology-preserving neighborhood of the SOMs also provides the ability to identify potential alternative spatial elements and temporal sequences where exact matching cannot be found by the predictive module.

Such an spatiotemporal unsupervised learning algorithm has numerous applications in the monitoring and anomaly detection space, which has inherently spatiotemporal data such as video surveillance [6], patient monitoring [7] and agriculture monitoring [8]. Consider video surveillance for anomaly detection as an example application. In this context, each image in the video represents a spatial element, and the complete video can be regarded as spatiotemporal data. Depending on the incident of interest, the length of the video segments may vary, and we need to consider variable-length sequences. Since each image can exhibit high variability, it is essential to abstract them when considering the sequences, highlighting the significance of separating spatial and temporal elements. Anomalies can vary depending on the scenario, emphasizing the importance of learning the data’s structure through unsupervised learning. Moreover, the anomaly scenarios can change dynamically, requiring continuous updating of the prediction with incoming data.

The rest of the paper is organized as follows. Section 2 reports the current literature in spatiotemporal learning and prediction, specifically, self-organizing maps (SOMs), hierarchical temporal memory (HTM), and vector symbolic architecture (VSA). The proposed algorithm, a two-layer self-organizing map with vector symbolic architecture for spatiotemporal sequence learning and prediction, is described in Section 3. Section 4 presents the empirical evaluation of this algorithm using two benchmark datasets and three real-world datasets. Section 5 concludes the paper.

## 2. Related Work

Most existing research focuses on supervised learning methods to achieve spatiotemporal pattern recognition [9,10]. Researchers use these methods to explore deep learning approaches to achieve spatiotemporal predictions; however, they attempt to use nature inspired unsupervised approaches to learn spatiotemporal patterns [11], and in this research we focused on those approaches.

Most of the current research in spatiotemporal pattern recognition focuses on the use of supervised learning methods, as explored in studies such as [9,10]. These methods utilize deep learning techniques to make spatiotemporal predictions. However, there have been attempts to explore nature-inspired unsupervised approaches for learning spatiotemporal patterns [11]. Our focus in this study was on these unsupervised approaches.

Research on unsupervised learning for spatiotemporal pattern detection and prediction has focused on identifying unforeseen structures in data considering both spatial and temporal attributes [12]. A number of time-series data clustering techniques have been proposed that include hierarchical clustering, density-based clustering, and model-based clustering in the fields of computer vision [13], human behavior [14], speech processing [15], and geo data [16]. But none of the existing clustering techniques accommodate all 4 key requirements expect from machine learning algorithms for fast-changing spatiotemporal patterns.

Hierarchical and multi-step clustering methods develop representations of the inherent structure of the data in multiple steps of unsupervised learning with generally increasing levels of granularity. Hierarchical aligned cluster analysis (HACA) is one such method that uses motion primitive clustering followed by dynamic alignment to cluster human activities [13,17]. These methods cluster action primitives and use temporal associations to determine the grouping of human actions. Furthermore, ST-DBSCAN [12] and ST-OPTICS [18] are two algorithms that use density-based techniques to group data into spatiotemporal clusters. These density-based algorithms consider both spatial and non-spatial features by accommodating two distances for spatial and non-spatial data to generate comprehensive clusters that represent spatiality and temporality. Both hierarchical and density-based clustering techniques identified above consider spatial and temporal aspects as two input dimensions in generating clusters but do not develop separate spatial and temporal output representations. Furthermore, these techniques do not attempt to address the need of handling variable length in sequences [19].

In the rest of the section, key concepts of the new algorithm that we proposed to address the requirements are described.

### 2.1. Self Organization and Self-Organizing Maps

Self-organization is a process where a form of overall order arises from local interactions between parts of an initially disordered system. The self-organization process is spontaneous when sufficient energy is available, without any need for control by any external agents [20]. Self-organizing maps (SOMs) represent a computational recreation of this process that projects a nonlinear high-dimensional input space into a reduced dimensional discretized representation, called a feature map, while preserving the topological relationship in the input space [5,21]. As input signals are presented, neurons compete amongst each other based on Hebbian Learning [22] for ownership of the input, and the winner strengthens its relationships with this input. Since the early 1980s, the SOM has been a popular unsupervised neural network model widely used for exploratory data analysis, visualization, and clustering. Compared to the other clustering algorithms such as K-means, the SOM produces a topology and neighborhood preserving low dimensional mapping that can be used as a discretized representation of spatial elements of the input space. The Integer SOM (iSOM) is one variation of the SOM algorithm, which is optimized for the hyperdimensional vectors [23]. iSOM is initialized with bipolar vectors and uses the cosine distance as the distance measure, which enables a resource-efficient alternative to the SOM algorithm. iSOM only utilizes integer operations, which makes it favorable for hardware implementations such as field-programmable gate array (FPGA). Even though SOM is widely used as an unsupervised learning technique, in its original form, it has several key limitations for representing spatiotemporal data; (a) it does not cater to the representation of temporal characteristics in data and (b) it requires the input data to be presented in a fixed number of features.

Due to these limitations, researchers have proposed extensions to the SOM to enable the representation of temporal patterns [24]. For example, Du et al. proposed a deep-stacked SOM architecture that captures spatiotemporal data in multiple layers to filter the background from images and detect dynamic objects in complex scenarios [25]. This method generates representations with incremental abstraction as deep layers. Gowgi and Garani proposed a mathematical theory of spatiotemporal self-organization based on ideas on reaction-diffusion [24]. Nawaratne et al. proposed a method to embed contextual information for the spatial vector using a fixed length window to cluster spatiotemporal data within SOM based structures [26]. But none of these techniques develop separate representations of the spatial and temporal elements and therefore do not have the flexibility of building and adapting models for frequently changing data in dynamic environments [27].

A hierarchical SOM method has been proposed based on two Kangas maps [28] representing spatial and temporal aspects, which are then merged to generate a spatiotemporal representation [29]. This method considers the spatial structure and temporal structure separately, but these independent maps do not preserve the relationship between spatial and temporal information, which is important to predict the future elements of a sequence. Even though the SOM structure generation is unsupervised, by adding a learning layer to maintain a memory of labeled data (when labeled data are available), SOM can be used for the classification tasks [30,31,32]. With such labeled data, it is also possible to identify the uncertainty of the data [33]. Therefore, SOM can be used for satisfying both the unsupervised learning requirement as well as carry out predictions based on past learning.

### 2.2. Hierarchical Temporal Memory (HTM)

HTM is a brain-inspired spatiotemporal learning algorithm. HTM can continuously predict the next element of a sequence and can also handle variable-length sequences. When considering the related research, HTM satisfies most of the algorithmic requirements identified above, including the separated representation of spatial and temporal aspects in the data. However, HTM requires inputs to be converted using a special encoding to sparsely distributed vectors in higher dimensions to enable continuous prediction. Generating such a representation with frequently changing data can be difficult, especially when near real-time predictions are important. HTM also utilizes the same computational architecture to model spatial and temporal elements, where cortical columns and mini-columns represent spatiality and cross-columnar lateral connections represent temporality. Although an elegant computational model, this architecture needs to create and strengthen cross-columnar connections to capture patterns from input data; therefore, it requires a longer processing time and also slow adaptation to changes, which is a disadvantage with fast and frequently changing data. Furthermore, due to higher dimensional data usage computational complexity is high when using the Von Neumann architecture [34]. There are a number of attempts to transfer the HTM algorithm to the spiking domain to improve the performance [35,36]. Further, there are different hardware implementations to improve the performance [37].

Inspired by the human neocortex, HTM uses columnar architecture to build a spatiotemporal structure to capture spatiotemporal data. The columnar structure has two layers called the spatial pooler and temporal pooler. The spatial pooler extracts spatial information of the underlying data and the temporal pooler enables sequence memory. HTM requires input data to be transformed into sparse distributed representations (SDR) prior to being fed into the two main components. According to the inventors of HTM, SDRs are inspired by the observation that activity patterns in the biological neocortex are very sparse, with less than 2% of neurones being active in a particular instance. SDRs have similar properties as HD vectors and utilize similar mathematical operations [38].

The spatial pooler receives hyperdimensional (HD) vectors as input and spatially structures the data points into HD vectors using a grid of neurones that has adaptive weights. The output of this spatial structure is a sparse distributed vector (SDR) where similar vectors result in similar patterns. The sequence memory capability is the second component of the HTM algorithm that is used to capture the context and temporality of sequences. This context and temporality are used to predict the next element of a sequence. The features of the spatial pooler are analogous to the SOM algorithm, but the SOM algorithm is more robust compared to the HTM one. In this work, we explore this behavior to develop a novel spatiotemporal algorithm that is capable of representing fast-changing dynamic spatiotemporal patterns.

### 2.3. Vector Symbolic Architecture (VSA)

VSA, also known as Hyperdimensional Computing (HD) [39], is a bio-inspired method for representing concepts and their meaning and enable cognitive reasoning through simple operations [40]. VSA uses hyperdimensional vectors to represent information in a distributed or holographic manner [41]. VSA has been successfully used to represent variable-length sequences and identify the structure of sequences [42,43]. VSA is based on the idea of distributed data representation. All the representations are in very high-dimensional vectors, where vector representations are distributed. This differs from the VSA from traditional localist data representation, where a single bit or group of bits represents a meaning without reference to the other bits, in contrast with VSA in which only the total set of bits (vector) can be interpreted.

The hyperdimensionality of the vector space is fixed for a single application, which is a hyperparameter for the application. The representational space for the given application caters to all the individual concepts in the application using hyperdimensional vectors (HD vectors). HD vectors need to have the property of randomness, which is ensured by 0.5 normalized Hamming distance between any two vectors [39,41]. Depending on the numeric representation of the vectors, there could be a number of different encodings, including real numbers [44], complex numbers [45], binary [39], and bipolar [46].

Three operations are defined for the HD vector manipulation in VSA as bundling (+), binding (⊙), and permutation (p()). Bundling (+) is used to join multiple HD vectors and create new vectors that represent all the vectors joined. Binding (⊙) is used to generate new vectors from individual HD vectors, which are different from the initial vectors. Permutation (p()) is used for joining the sequence of vectors to create new vectors considering the order and individual features.

In the proposed two later SOM approaches, VSA was harnessed to address two key requirements for spatiotemporal sequence representation and prediction. Real-life spatiotemporal sequences can be of varying lengths, the original SOM does not cater to this need, and VSA was used as a bio-inspired solution by transforming the variable-length sequences to fixed length HD vectors. HD representation space is identified as an item memory for the application, which represents the relationship between identified hyperdimensional vectors and concepts (spatial elements captured by the original SOM). This item memory can be used to identify clean concepts from noisy HD representations. All the concepts are identified as the VSA alphabet, and the item memory feature is used to facilitate continuous online prediction of the next element of a sequence.

## 3. The Proposed Algorithm: ST-SOM

This section presents the two-layer self-organizing map algorithm with vector symbolic architecture for spatiotemporal sequence learning and prediction. The algorithm is called the ST-SOM, in reference to its spatiotemporality based on the SOM algorithm. Its architecture is made up of four components (Figure 1). The two layers of SOMs provide unsupervised representational learning capability while developing separate and independent representations of spatial and temporal aspects in the input. The VSA encoding component addresses the need for managing variable length in real-life sequences. Finally, the memory component enables the prediction of the next element of a sequence.

As shown in Figure 1, Spatiotemporal SOM accepts an input of variable-length sequences of spatiotemporal data $U_{T}$, which has *T* elements (T>=1). Each element $U_{T}\left(t\right)$ (t⩽T) is a vector of dimension *d* (d>0). The first SOM consists of N×N nodes with weight vectors of the same dimension as the input data and accepts individual elements from input sequences. The time dimension in the spatiotemporal data is not considered in the first SOM, which will generate a topology (neighborhood)-preserving mapping of spatial elements in input sequences. The second layer of SOM nodes generates an unsupervised representational learning of the complete sequences as inputs including time as a dimension. The transformation of variable-length sequences to fixed dimensionality of the second SOM weight vectors is achieved through the encoding module. The encoding uses the spatial elements identified in the first SOM and performs VSA sequence flattening considering the sequence order as per Equation (Equation 2). VSA encoding necessitates the second SOM dimensions to be hyperdimensional.

To facilitate the prediction of the next element of a sequence, the memory module is introduced to establish an association between sequences represented in the second SOM layer and spatial elements represented in the first SOM. When labeled data are available, the memory module will record labels for each node in the first SOM layer, which can be used to predict the class labels for temporal sequences.

The detailed steps of the proposed ST-SOM algorithm follows the Algorithm 1. The input for the algorithm consists of variable-length sequences ($\left\{u_{T}\right\}$ here, {} represent all sequences in dataset)) as well as two SOM layers, learning rates, and SOM map sizes (*N*). The algorithm expects sequences to be in multivariate format, and the domain-specific function Data_dependent_element_encoding() is used to convert sequences into multivariate sequences ($\left\{U_{T}\right\}$) for the ST-SOM algorithm.

Starting from step 2, the ST-SOM algorithm incorporates the learning mechanism. It begins with the first layer SOM, where the function Initialise_SOM() initializes the SOM map similar to the classical SOM algorithm [21], and the function SOM_learning_memory() performs the learning function of the SOM algorithm. In addition to the normal learning step, it also maintains the memory of the next element or output label predictions.
**Algorithm 1:** ST-SOM  **Input**: $\left\{u_{T}\right\}$, $size^{first\_layer\_SOM},lr^{first\_layer\_SOM}$,      $size^{second\_layer\_SOM},lr^{second\_layer\_SOM}$  **Output**: spatio−temporalstructure**1** $\left\{U_{T}\right\},d\leftarrow Data\_dependent\_element\_encoding\left(\left\{u_{T}\right\}\right)$**2** Initialise_first_layer_SOM←  
$Initialise\_SOM(size^{first\_layer\_SOM},lr^{first\_layer\_SOM},d)$;**3** $First\_layer\_SOM\leftarrow SOM\_learning\_memory(initialise\_first\_layer\_SOM,\left\{U_{T}\right\})$;**4** $\left\{U_{T}^{first\_layer\_SOM}\right\}\leftarrow BMU\_seq\left(\left\{U_{T}\right\}\right)$;**5** $\left\{U_{HD}\right\},D\leftarrow VSA\_encoding\left(\left\{U_{T}^{first\_layer\_SOM}\right\}\right)$;**6** Initial_second_layer_SOM←  
$Initialise\_SOM(size^{second\_layer\_SOM},lr^{second\_layer\_SOM},D)$;**7** Second_layer_SOM←  
$iSOM\_learning\_memory(initialise\_second\_layer\_SOM,\left\{U_{HD}\right\})$;

Before moving on to the second layer SOM, there is VSA encoding based on the first layer SOM Best Matching Units (BMUs), which are represented in steps 4 and 5 of the algorithm. Here, the function BMU_seq() is used to give abstract sequences where each element is represented by BMU of the first layer SOM ($\left\{U_{T}^{first\_layer\_SOM}\right\}$), and the function VSA_encoding() is used to convert each sequence into an HD representation according to Equation (Equation 2).

The last two steps cover the second layer SOM learning, which is similar to first layer but uses the IntegerSOM (iSOM) algorithm to support HD representation, and this second layer is independent of the input data formats. Each layer’s learning has to be sequential as the second layer’s learning depends on the first layer’s spatial representation.

### 3.1. SOM-Based Representational Learning of Spatial and Temporal Patterns

The two self-organising layers facilitate the separated representational learning of spatial aspects and temporal relationships inspired by findings in neuroscience. The SOM layers provides an added advantage over other clustering algorithms by capturing a neighborhood-preserving mapping of the input data where the neighborhood can be used as alternate spatial possibilities when predicting the next element of a sequence.

The first self-organising layer is composed of 2D SOM, where each side has *N* neurons, which result in N×N total neurons. Neurons in the first layer have the same dimension *d* as input vectors $U_{T}\left(t\right)$. The distance measure of the algorithm will change from cosine to euclidean depending on the data format. The first SOM functions as a spatial element identifier for the creation of an alphabet of HD vectors (VSA) to be used as item memory for VSA encoding. The number of nodes in the first layer neurons determines the size of the VSA alphabet where the elements of the alphabet are used to represent temporal sequences in the second layer.

The second layer SOM is independent of the input data format and therefore not impacted by varying lengths of sequences. It will always use HD vectors, and in the proposed implementation it uses the iSOM variation of the classical SOM algorithm [23]. The purpose of the second layer SOM is to identify the temporal structure of the sequences $U_{T}$. iSOM variation uses the cosine similarity for the distance measure and is presented as Equation (Equation 1) for the distance update where $f_{k}\left(\right)$ is a clipping function with |k| as the maximum value and $p_{i}$ is a vector of random numbers that are in range [0,1] with the same dimension as input vectors *D*; here, α(t) is the learning probability at each iteration *t*. $W_{ji}\left(u+1\right)$ is the weight of the $j^{th}$ weight vector $i^{th}$ position, and $W_{ji}\left(u\right)$ is the same weight value before the weight update. $x_{i}$ is the $i^{th}$ position input value.
(1)$$W_{ji}\left(u+1\right)=\left\{\begin{array}{ll} f_{\kappa }\left(W_{ji}\left(u\right)+x_{i}\right) & \mathrm{if}\;p_{i}\leq \alpha \left(t\right)\\ W_{ji}\left(u\right) & \mathrm{otherwise} \end{array}\right.$$

### 3.2. Hyperdimensional Encoding

This encoding process consists of three steps:Step 1: The identification of the VSA alphabet;Step 2: The creation of the item memory;Step 3: The conversion of the temporal sequence to VSA vectors.

Step 1 of the encoding process uses the nodes of the first SOM to link spatial elements in the input data to the VSA alphabet, where the N×N individual nodes are used to denote separate elements of the alphabet. This alphabet will therefore represent all the spatial concepts (elements) in the particular environment. Step 2 creates the item memory using the identified elements of alphabet. Item memory maintains the relationship between alphabet elements and related HD vectors ($h_{i}$, where 0⩽i<N×N). HD vectors are generated adhering to the VSA properties described in Section 2.3. For the implementation described in this article, individual HD vectors of dimension *D* were generated by random initialization using the bipolar encoding and stored in an item memory for encoding sequences.

Step 3 of encoding converts the variable-length sequence $U_{T}$ to *D* dimension vectors. In each time epoch *t*, the input element $U_{T}\left(t\right)$ is passed through the first SOM layer to identify the closest matching SOM node. The corresponding VSA alphabet element in item memory is then identified using the SOM node.

Temporal sequences are represented using a single HD vector of fixed *D* dimension using the bundling and permutation operations on individual alphabet elements to generate the VSA representation of a sequence. This procedure is called flattening or levelling the sequence [39]. A bundling operation is used to aggregate the individual elements into a single HD vector. A permutation operation is used to enforce the order of the elements in the sequence.

Equation (Equation 2) shows the flattening operation using bundling (+) and permutation (p()) for a variable-length sequence of G elements. $H_{i}$ is generated HD vector for element *i*.
(2)$$U_{\mathrm{HD}}=h_{1}+p\left(h_{2}\right)+p^{2}\left(h_{3}\right)+\ldots p^{G-1}\left(h_{G}\right)$$

Figure 2 shows the sample hyper-dimensional encoding of the text sequence considering the sentence “Sydney Opera House is an Australian landmark”. The first step of separating the sentence into words and the identification of word embedding is not depicted in the figure. Individual words are considered as U(t) in this example and passed to the first layer SOM to position the individual words as a spatial or invariant representation. Due to the neighborhood preserving topological mapping property of the SOM, each word will be positioned among words with similar meanings or context from the inputs received.

The item memory component will link the sequence (sentence) to generate a sequence of HD vectors of fixed dimension *D*. The HD vectors representing individual elements (words) of the sequence are then converted to a single HD vector of dimension *D* using the Equation (Equation 2).

### 3.3. Memory Module and Prediction

The memory module is introduced to the proposed algorithm based on theories of memory and learning [47], to facilitate the linking of spatial elements represented in the first SOM layer with the related sequences captured in the second SOM layer. Maintaining the associative memory enables the prediction of the next element (occurrence) of a sequence based on spatial elements in the first-layer SOM. Further, keeping track of the labeled data with the second-layer sequence structure makes it possible to identify (classify) a sequence using individual elements.

The memory module uses an associative memory to keep track of the labels for each node within the second self-organising layer. A similar approach of using associative memory to identify the uncertainty in predictions was used in [33] that uses similarity-based pattern recognition [48].

When a direct match for an element is not present in memory, the module will search the neighborhood within the SOM to find an appropriate next element. If the neighborhood does not provide an alternate element, the next element is predicted considering the last *n* elements. If data are labeled, a similar approach is followed to generate the labels at the first layer SOM. The Algorithm 2 follows the steps for generating labels from memory considering the SOM neighborhood.

In the algorithm, generate() is a method that generates a new label from a given set of labels. The implementation of the generate() depends on the label structure. If labels are numerical, then this function can be the average of the given labels. neighborhood provides the neighborhood clusters, and average_size gives the labels of last *n* elements.
**Algorithm 2:**  Predict label from memory

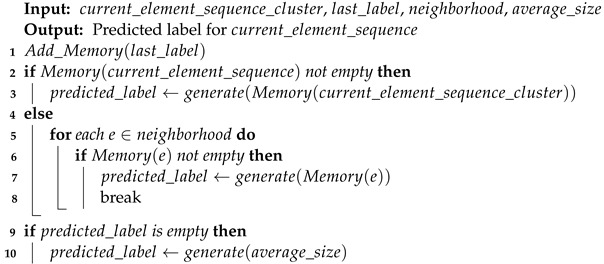


## 4. Experiments

This section presents a series of experiments to demonstrate the functionality of the two Self-Organizing Map (SOM) layers in learning spatiality and temporality in data, as well as the predictive capability of the complete ST-SOM algorithm. Section 4.1 showcases the 2D visualization capability of the proposed ST-SOM algorithm. Additionally, we evaluated the classification accuracy and ST-SOM’s ability to achieve comparable results to supervised algorithms. In Section 4.2, we compared the state-of-the-art HTM algorithm and found that it achieved higher predictive capability for time-series data and could obtain an error reduction of 45%.

### 4.1. Representational Learning of Spatial Elements and Temporal Relations

Experiments in this section use benchmark multivariate variable sequence data set in the video domain and video-clustering-based action recognition using the KTH dataset. The objective of the experiments is to demonstrate the unsupervised representational learning of spatial elements with the first SOM layer and the capture of temporal relationships as sequences by the second SOM layer. Finally, the classification accuracy of the data set is evaluated to demonstrate the predictive capability using the memory module.

#### 4.1.1. Representational Learning of Spatial Elements with the First SOM Layer

KTH [49] is a video action recognition dataset that contains videos of six activities (walking, jogging, running, boxing, hand waving, and hand clapping) performed by 25 subjects. The actions are captured in four different scenarios: outdoors, outdoors with scale variation, outdoors with different clothes, and indoors. The dataset contains 599 videos comprising 289,715 frames and 11,375.32 s captured at 25 fps and 160 × 120 pixels.

Videos are sequences of images (frames) that are represented as a multivariate vector. The videos have variable lengths and as such are made up of variable-length sequences of image frames.

We have used 2D-poses as the individual image data similar to the method introduced in [50]. The feature vector for the individual image in a video is generated as follows. First, the 2D human pose is extracted using the deep learning OpenPose framework [51]. This extracts 25 key points of the body, which are then used to extract 14 angles from the human body joints. These 14 angles are used as the feature vector representing the respective image. To reduce the length of the vector and avoid repetition, these 14 angles are extracted from every $n^{th}$ consecutive frame from the video. The value *n* is selected empirically with n=10 for the described experiment.

The first SOM layer uses the 14 feature vector values to generate a self-organized map representing the images. This self-organized structure provides an abstract grouping of the human body movement from the images based on the 2D poses, which represent the spatial elements (or the spatial alphabet). The optimal learning rate for the SOM is identified empirically. Figure 3 shows the cluster accuracy of the final temporal cluster results with the first SOM layer learning rate and normalized mutual information (NMI). According to Figure 3, the highest accuracy is achieved with a learning rate of 0.3. For this experiment, the number of iterations has been kept at 100.

To demonstrate the representational learning of spatial clusters by the first SOM, which is considered as the alphabet of elements with which sequences are made up, three separate sequences (videos) representing hand-clapping, jogging, and running classes are compared with the first SOM layer. Figure 4 shows a selected sample of the first layer SOM clusters (nodes), which represent elements of the identified alphabet and the first five representative images for each element from each cluster. The elements (clusters of poses) are labeled as A to L to represent the elements captured by the first SOM. For example, Figure 4a shows fove spatial elements made up of different poses from the sequence of hand clapping, captured and represented in the first SOM. The sample for each alphabet element selected (columns 1–5) shows that the same pose (alphabet element) by different individuals and in different conditions such as lighting is grouped in the same alphabet element. It can also be seen that elements (or alphabet) making up hand-clapping are fully separated from jogging and running. Jogging and running are shown to have a significant overlap of elements, which can be intuitively understood since these two activities can have much in common in terms of movement. In further analysis of the alphabet elements, it can be seen that sequences of jogging and running have similar leg movements, which results in the overlap in elements while differing in hand movements. This is reflected in the element (cluster) overlap; both jogging and running have *F*, empty image elements, and *G* and *H* running image elements as overlapping. This output can be further interpreted by observing the second SOM feature map for the related sequences in Figure 5, where hand-clapping sequences are positioned further away from the jogging and running sequences. This experiment demonstrates how the first SOM layer automatically captures the spatial elements of the data with which the sequences in the second layer are made up, therefore generating an alphabet of building blocks of the sequences.

Next, HD vectors are created according to Equation (Equation 2) for the video trajectory. The second self-organizing layer clusters the videos, and Figure 5 shows the SOM 2D map of the KTH dataset. According to the 2D SOM map of the second SOM layer, hand movement activities (boxing, hand-clapping, and hand-waving) and activities with the movement of legs (jogging, running, and walking) have clearly separated into two regions, with individual activities further separating within the main regions.

The SOM algorithm uses topological presevation to create a spatiotemporal representation of data. To determine the topology preserving capability of the two SOM layers, we analyzed the topographical error [52]. This error indicates when the data samples with BMU1 and BMU2 are not adjacent. A map with topological preservation should have lower values. For the KTH video dataset, the first layer had a topographical error of 0.16, while the second layer had an error of 0.05. The second layer had better topography compared to the first layer. This was expected because the first layer had a higher distribution of action images. However, in the second layer, the error was further reduced due to the use of the first layer output, creating an abstract spatiotemporal representation.

In this experiment, we evaluate the algorithm using classification accuracy, where the Spatio-Temporal-SOM algorithm predicts the video class labels using the memory module following the Algorithm 2, where we were able to achieve 86.6% accuracy. To highlight the innovation and benchmarking purposes, the proposed method is compared with two other comparable methods (Yang et al. [53] and Peng et al. [54]), which have certain similarities. Accuracy comparison is available in Table 1. Even though the proposed method does not have the highest accuracy, it has comparable accuracy with the other two methods; further, the new method has additional advantages, as discussed in the comparison below.

Yang et al. [53]’s method is similar to the proposed method in terms of the two layers used to cluster the videos where the first layer is used to capture motion primitives and the second layer to capture the relationship of motion primitives using either statistical distribution or string comparison. The first layer of this method is similar to the proposed method where motion primitive is identified using clustering, which is similar to the SOM identification of the alphabet, but the second layer in the proposed method is more generic and could be used for multiple applications and use cases such as clustering and prediction. In contrast, Yang et al.’s method of statistical distribution does not consider the order of primitives, and the string comparison method is not scalable and does not operate on partial sequence matches.

Peng et al. [54]’s method uses multiple view representation to combine auxiliary information in the video to build a rich representation to represent the video and generate clusters. This is achieved by combining the scene information with motion information. Even though this method uses a multi-layer method where the first layer identifies different information to represent the video and the second layer combines that information to generate a full representation of the video, it does not consider the individual primitives in the video frames and their trajectories and does not provide the alphabet building blocks as the proposed spatiotemporal SOM, which could be used as generic building blocks.

#### 4.1.2. Representational Learning of Temporal Sequences with the Second SOM Layer: KTH Action Recognition

The following demonstration uses the spatial elements captured in the Section 4.1 experiment clustering results and illustrates the sequence learning by the proposed algorithm using the “handclapping” class in the KTH video dataset. The objective of this demonstration is to show how the spatial elements from the first SOM make up the different temporal sequences captured by the second SOM. The presence of distinct elements, their order, and their repetition in a sequence can be used to identify similarities or differences in patterns, individual peculiarities in activities, and for predicting next occurrences. To identify similar sequences and group them into clusters, a pairwise alignment technique from bioinformatics was utilized [55]. The pairwise alignment was extended to allow partial matches in similarity calculation to accommodate SOM neighborhoods. Refer to Section A.1 for the implementation of the extended pairwise alignment method.

First, different cluster nodes in the second layer SOM were identified for the handclapping class. For each node representing the sequence for handclapping in the second SOM layer, a pair of instances in that cluster node is passed through the modified sequence alignment method and recurring aligned patterns are extracted.

Table 2 shows the identified recurring sequence patterns in four nodes in the hand clapping class from the second SOM (Figure 5), where node IDs from the first SOM are used to represent the spatial elements. The node positions from the second SOM ((0,19), (0,18), (0,12), (3,15)) were selected to show the variation within the single class of handclapping and how the spatial elements have been used in the second SOM sequences to capture such variation. Figure 5 provides a visualization of selected sequences for the identified sequence patterns.

Figure 6 was generated to further investigate the selected sub-patterns and relate and interpret the variation in these sequence patterns using the first SOM layer spatial elements. Two representative patterns from each of the four nodes were selected. With Figure 6 it can be seen that all four patterns have the three elements 542 (hands together in clapping position), 576 (hands apart and held high), and 668 (hands apart and held low). (0,19) and (0,18) nodes are adjacent in the second SOM, and the represented sub-pattern sequences are only different due to the longer period hands that are held apart in (0,19). (0,12) is in the same neighborhood as (0,19) and (0,18), and the representative sequence is still made up of elements 542 and 576 in a different combination. Node (3,15) is further away compared to the other three nodes, and it can be seen that the pattern includes a different element 668 in a sequence pattern with 542.

#### 4.1.3. Representational Learning of Temporal Sequences with the Second SOM Layer: Indoor Movement Sensor Data

The next experiment is based on the radio signal strength (RSS) data to predict indoor movement. In this application, each point of user movement is captured using four Wireless Sensor network nodes, and complete movement from one of six pathways is a variable length trajectory of four RSS sensor values.

This dataset represents a benchmark for the area of Ambient-Assisted Living applications. The task is a binary classification task where class +1 is associated with location changing movements and −1 class with the location preserving movements. Data are collected during the user movement at a frequency of 8Hz. The measurement campaign involves three different environmental settings, each of which comprises two rooms separated by a corridor. Figure 7 shows a simplified illustration of the types of user trajectories considered. Dataset contains 314 sequences and 13,197 steps.

The next steps are similar to the previous experiment where HD vectors are generated for the abstract groups and trajectory HD vectors are generated to pass to the second self-organizing layer. The second level self-organizing layer outputs the clusters of the movement pathways.

Figure 8 shows the final cluster output of the indoor movement sequences. Even though there are no separate clusters for the two classes, multiple clusters can be identified for each class based on the distribution of the SOM nodes.

In this experiment, similar to the previous experiment classification accuracy is used to evaluate the proposed algorithm. Table 3 shows the accuracy comparison and Table 4 shows the parameters used for the ST-SOM algorithm. The algorithm predicts the class labels using the memory module following the Algorithm 2 and achieves 87.3% accuracy. The proposed method is compared with method [56], which has 89.5% accuracy as a supervised task. Therefore, the proposed method has accuracy on par with other supervised methods.

### 4.2. Demonstration of the Predictive Capability and Comparison with State of the Art

The previous experiments were designed and carried out to demonstrate the functionality and inner workings of the two SOM layers. In this section, the complete ST-SOM is applied to three real-life data sets to demonstrate the predictive capability based on the sequences learnt. The output of the ST-SOM is compared with the HTM algorithm since, as discussed previously, HTM satisfies several requirements highlighted as important from a spatiotemporal data-based real-time predictive algorithm. HTM uses an unsupervised learning paradigm and caters to variable-length sequences but does not architecturally support the separate learning of spatial and temporal aspects in the input. The following experiments highlight the advantage gained by ST-SOM over HTM due to the separated learning of spatiality and temporality, which is especially significant with real-time predictions.

The first dataset is based on the Taxi passenger demand dataset introduced in [57] to compare HTM with other predictive algorithms, where HTM is shown to produce state-of-the-art results. The dataset has aggregated taxi passenger counts in New York City for 30-min intervals. This real-world dataset exhibits rich patterns and contains concept drifts that are difficult to detect and learn with most sequence learning algorithms.

The second and third datasets are part of the NAB anomaly detection benchmark dataset [58]. The second dataset is from Amazon Web Services (AWS)’ monitoring CPU usage. The third dataset is the temperature sensor data of an internal component of a large, industrial machine. Both these datasets are similar in terms of features and underlying behaviors to the first dataset, which consists of a timestamp and value pair, and similar procedures have been used to predict the next element of a sequence.

This experiment demonstrates the functionality of the ST-SOM memory layer in addition to the two self-organizing layers to achieve continuous predictions based on spatiotemporal sequences. The memory module uses Algorithm 2 to predict the label value, and the generate() function is considered as the average of given labels as the prediction is a continuous numeric value. This experiment predicts the next value of the continuous sequence for the three datasets, and the Spatio-Temporal-SOM algorithm is compared with the HTM implementation as per the reference code provided in [57]. The first self-organising layer in ST-SOM uses the same scalar encoding used in [57] to cluster individual sequence elements, and the second level self-organizing layer clusters the continuous sequences with a moving window(*w*). The optimal moving window size is identified empirically, and for this experiment, w=50 is considered for all three datasets. Error values for both HTM and Spatio-Temporal-SOM algorithms are calculated according to the mean absolute percentage error (MAPE) metric, which is in Equation (Equation 3).
(3)$$MAPE=\frac{\sum_{t=1}^{N}\left|y_{t}-\hat{y_{t}}\right|}{\sum_{t=1}^{N}\left|y_{t}\right|}$$

Table 5 shows the final MAPE error values of the two algorithms for the three datasets. The Spatio-Temporal-SOM algorithm shows significantly lower final error values compared to the HTM algorithm for both the Taxi Passenger and CPU usage datasets. The error values for the machine temperature dataset are close, with the Spatio-Temporal-SOM error being slightly lower.

Figure 9 shows the change in the MAPE error value with the number of inputs for the HTM and ST-SOM algorithms for the New York taxi passenger dataset. According to the graph, the Spatio-Temporal-SOM algorithm starts with a higher error value as it has not been exposed to any of the patterns. When a significant proportion of the input data has been presented, the error value decreases. Since the HTM goes through a training/learning phase prior to the predictive phase, HTM errors start with a lower value.

As per the graph for the Taxi passenger demand data in Figure 9, there is a noticeable increase in error around the 10,000 inputs data point. This point corresponds to a concept drift in the data stream resulting from a significant change in input values. The Spatio-Temporal SOM algorithm recovers from this concept drift with a minor increase in the error compared to HTM.

Since the two-layer architecture of the ST-SOM results in capturing the spatial alphabet separately from the temporal sequences, the changed post-drift pattern could be constructed faster from the already learnt first-layer alphabet. In the HTM, the columnar architecture caters to the capture and representation of both the spatial and temporal elements. As mentioned in the related work section above, this architecture needs to create and strengthen cross-columnar connections to capture patterns from input data and therefore requires a longer processing time and slow adaptation to changes. To learn the post-drift changes, the architecture has to gradually adapt to new inputs, which takes much longer compared to the ST-SOM pattern learning mechanism. In contrast to ST-SOM, the HTM architecture consists of sequence pathways with weight values between mini-columns. To adapt to concept drift, weight values need to adapt using the Hebbian like learning and gradually change the sequence pathways. Making use of the first SOM layer, an analysis of the patterns was carried out to identify the potential causality of the concept drift. Two different main patterns were noted for weekdays and weekends, and during weekends there were high passenger counts around midnight and low in the morning (around 7 a.m. to 9 a.m.). Different sub-patterns of counts were noted for different regions. On the day of the concept drift (27 January 2015), the lowest values are seen to be lower than other normal lower counts, requiring a new separate representation in the first SOM alphabet. On further investigation, we found that there had been severe snowfall on this day [59]. The analysis and investigation further confirmed that the new spatiotemporal SOM can quickly adapt to changes and the separation of spatial and temporal representations. The new alphabet element can now be used when there is another such unusual occurrence, even if it happens on a different day and at a different time.

Figure 10 further confirms this behavior with two additional real-world datasets. The second dataset depicted in Figure 10 further confirms the ST-SOM ability to speedily adapt to changes. Although there is no sudden increase in error value, there is a gradual increase in error value around the 2500th input. The ST-SOM algorithm recovers from this error increase more efficiently and faster compared to HTM. The third dataset has lower error values compared to the other two datasets as per Figure 10b. But having a lower final error value further confirms the ST-SOM’s ability to speedily recover from sudden or gradual input value change in the continuous sequences, and therefore it is a better candidate for real-time prediction compared to HTM.

## 5. Conclusions and Future Work

The fast-changing and dynamic nature of the current information age is represented by volatile and frequently changing spatiotemporal patterns. Information and behaviors embedded in such data can provide valuable insights for decision-makers if captured accurately and in a timely manner. This paper proposes four key requirements that machine learning/AI algorithms must satisfy to provide practical value in such environments and data: (a) unsupervised learning ability for learning without pre-labeled data, (b) representational learning of spatial and temporal elements as separate building blocks for the construction of spatiotemporal patterns and adapt to changing environments, (c) the ability to represent and manipulate temporal sequences of variable length (order), and (d) the continuous online prediction of the next occurrence of a sequence based on the sequential recall of a pattern from memory.

In the literature, different algorithms try to address different features of fast-changing spatiotemporal patterns, but none of the existing algorithms capture all four of the key requirements identified above. The HTM algorithm closely relates to the key requirements for spatiotemporal patterns, but due to the implementation limitations and constraints, it is not able to adapt to the fast-changing spatiotemporal patterns. This article introduces ST-SOM as a new nature- and neuro-science-inspired algorithm that caters to these needs, demonstrating the functionality with five benchmark datasets for video, sensor, and timeseries data.

Despite being an unsupervised learning algorithm, the proposed ST-SOM algorithm is able to obtain in-par results for KTH and indoor movement datasets for accuracy. However, ST-SOM is not a simple prediction algorithm, and it achieves this by building the spatiotemporal structure of the data. To demonstrate the ST-SOM algorithm’s versatility, we have further explored the KTH dataset and demonstrated the clustering and 2D visual representation of spatiotemporal data. Similar image actions clustered in the first layer helped to develop a spatiotemporal 2D representation of the variable-length action videos, and the topographic error demonstrates the quality of the mapping generated from ST-SOM algorithm. The explainable visualization capability of the SOM algorithm [60,61] is leveraged in the ST-SOM algorithm to demonstrate the variable length spatiotemporal pattern learning behavior.

Furthermore, for time series data, ST-SOM was able to show higher accuracy compared to the HTM algorithm. With this comparison, we were able to demonstrate the fast-changing data adaptation capability of the proposed algorithm over HTM. The results of the experiment indicate that the new algorithm has the ability to learn spatial and temporal patterns separately, which enables it to adapt quickly and recover from concept drifts. However, it is important to note that this experiment only considered the scenario where fast-changing dynamic spatiotemporal patterns were involved, and individual spatial elements did not change. As a result, there was a static spatial alphabet and dynamic spatiotemporal patterns. To address the scenario where both spatial and spatiotemporal patterns are dynamic, we need to consider an adaptive neighborhood function and learning rate for the SOM algorithm. It is worth noting that this is a limitation of the current algorithm, and it will be explored further with the distributed SOM [62]. In addition, varying degrees of abstraction at the first layer may result in the dispersion of the spatiotemporal patterns distinguished in the second layer. For now, we have maintained a consistent level of abstraction. However, hierarchical clustering could be used to explore different levels of abstraction, which will be investigated in future research.

The ST-SOM algorithm has been proposed as a solution to demonstrate the four key features necessary for handling fast-changing spatiotemporal patterns. However, as it inherits the characteristics of the SOM algorithm, it requires parameters for learning rate, map size, and the number of iterations for both spatial and temporal layers. As the purpose of this paper is to introduce a new algorithm, for the convergence of the two SOM layers, we have kept the number of iterations to a reasonably lower number and evaluated the effect of the learning rate. However, this requires further investigation and sensitivity for SOM parameters and will be further explored in future work. Additionally, to showcase the full capabilities of the ST-SOM algorithm, experimentation with real-world datasets is necessary.

## Figures and Tables

**Figure 1 biomimetics-09-00175-f001:**
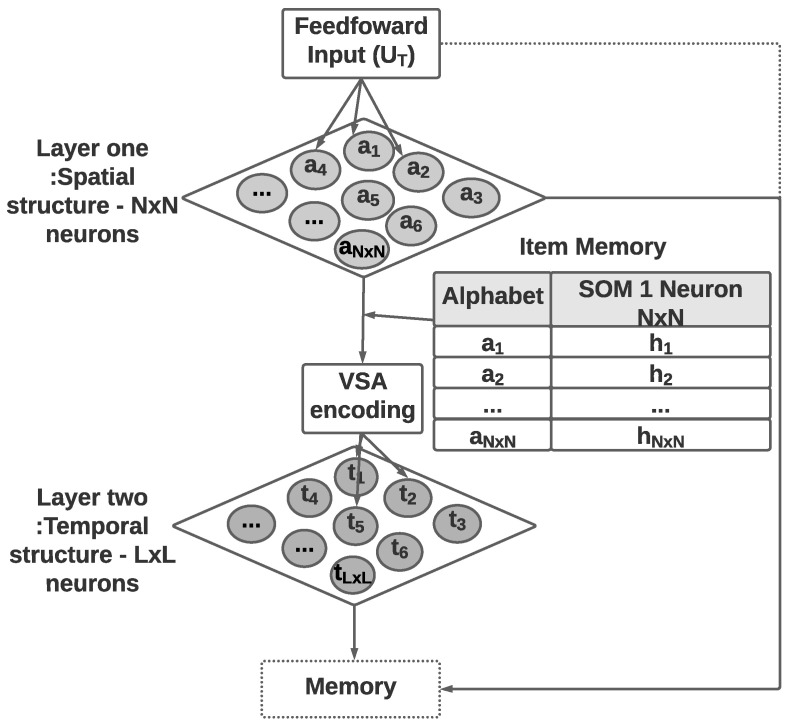
The architecture of the Spatio-Temporal-SOM algorithm with two separate SOM layers for representing spatial and temporal structure, an encoding module to transform input data to HD vectors, and a memory module to retain past sequences and enable prediction of next elements.

**Figure 2 biomimetics-09-00175-f002:**
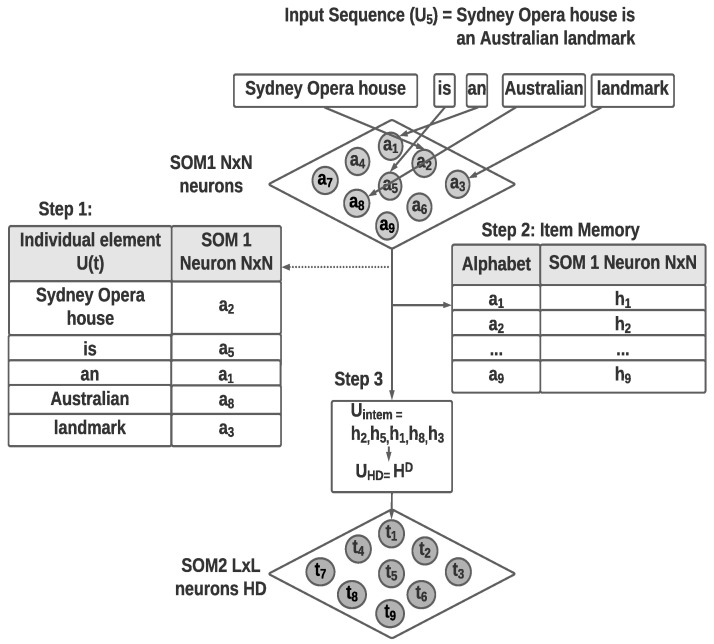
Example VSA encoding of the sentence “Sydney Opera House is an Australian landmark”: Step 1: use first SOM layer to identify nodes representing individual elements, Step 2: identify VSA alphabet using layer one SOM nodes, and Step 3: Convert the temporal sequence to a VSA vector.

**Figure 3 biomimetics-09-00175-f003:**
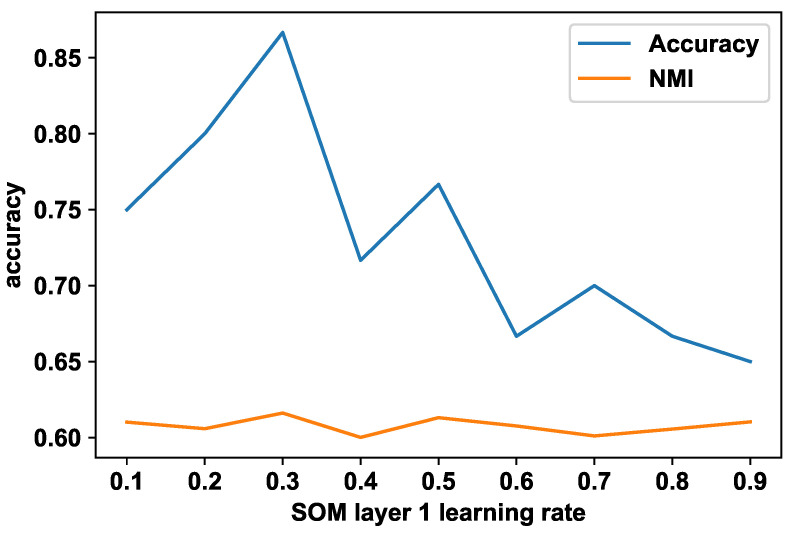
KTH action video cluster accuracy with first-layer SOM abstractions using Spatio-Temporal-SOM.

**Figure 4 biomimetics-09-00175-f004:**
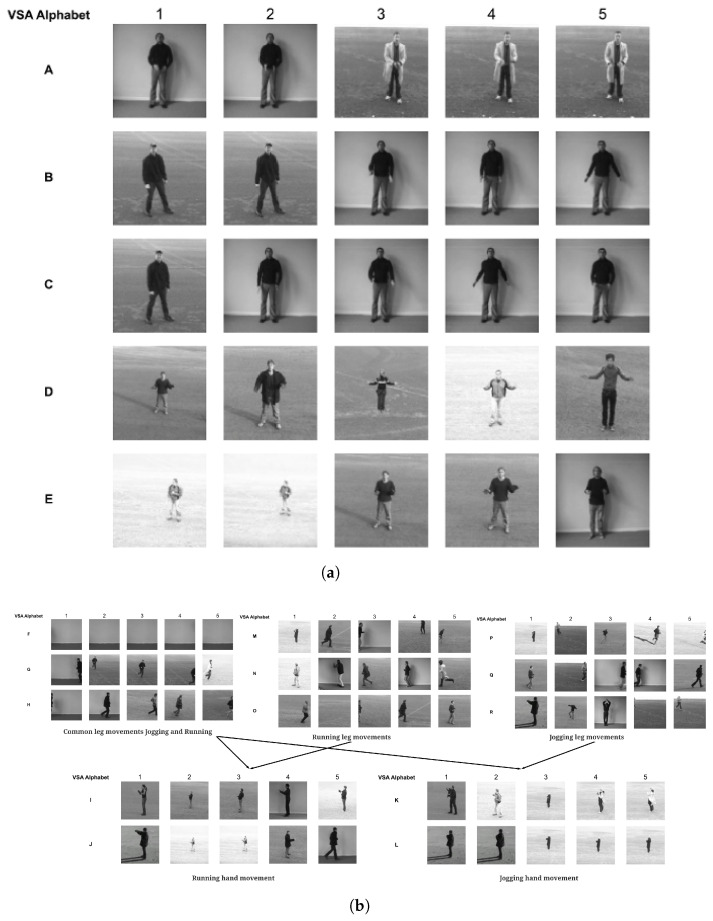
A sample of spatial elements captured by first SOM layer for KTH data, where hand-clapping is clearly separated from jogging and running (**a**) vs. (**b**). Jogging and running can be separated into five segments (**b**), where 1^*st*^ segment identifies the jogging and running common leg movements, the 2^*nd*^ and 3^*rd*^ segments identify separated leg movements of jogging and running, and the 4^*th*^ and 5^*th*^ segments identify separated hand movements. (**a**) Elements of hand-clapping. (**b**) Elements of jogging and running.

**Figure 5 biomimetics-09-00175-f005:**
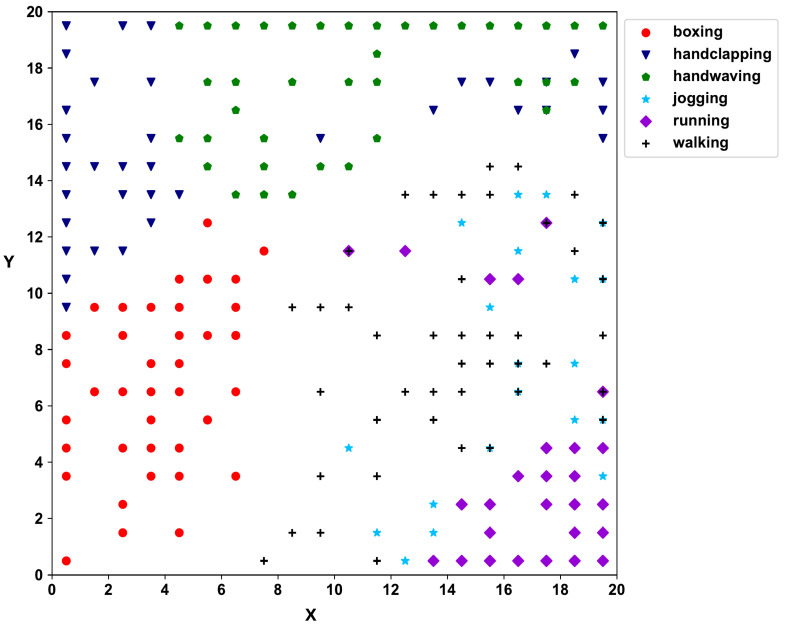
The SOM map for the second layer of KTH data showing groupings of sequences.

**Figure 6 biomimetics-09-00175-f006:**
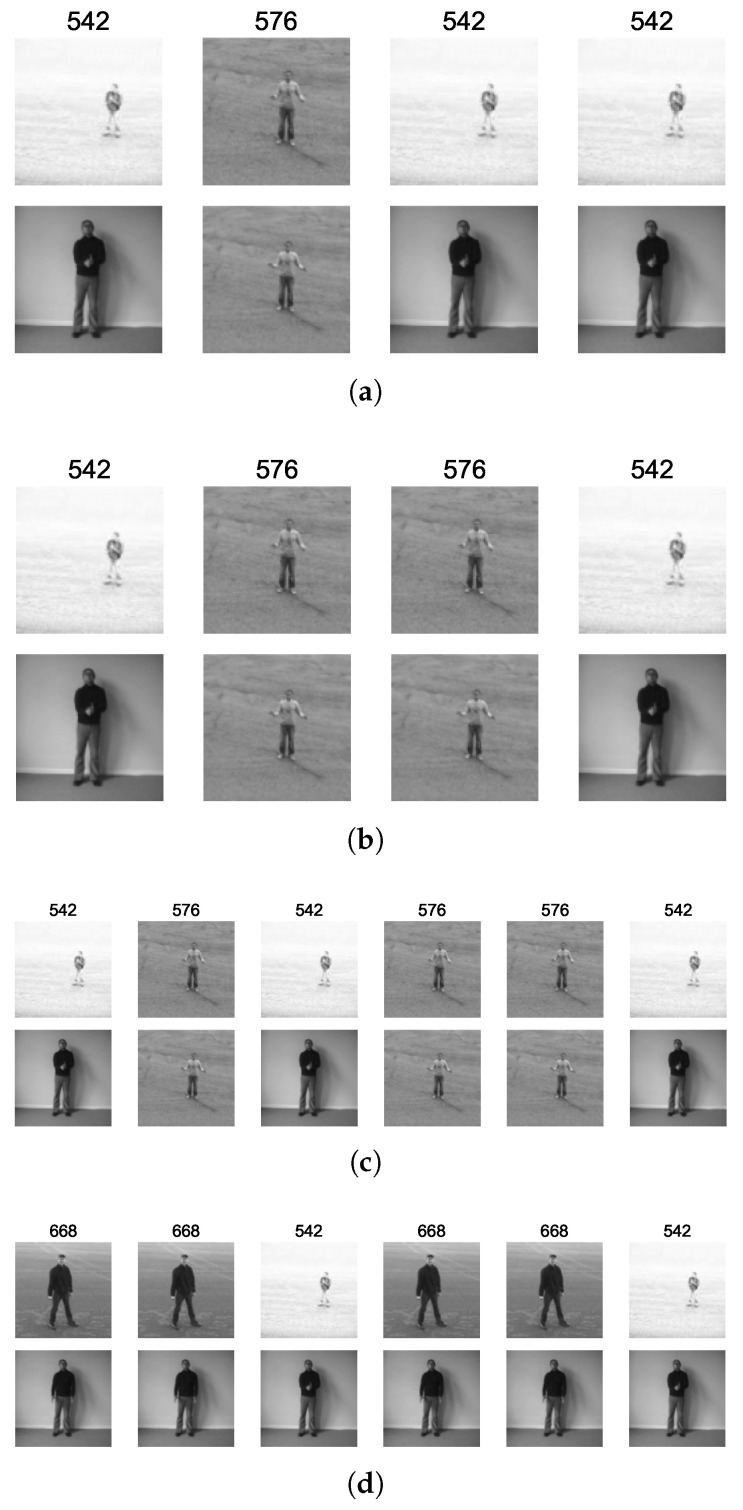
Recurring sub patterns in hand-clapping in four nodes (0, 19), (0, 18), (0, 12) and (3, 15); each node is showing first two images of the node. (**a**) Hand-clapping (0, 19). (**b**) Hand-clapping (0, 18). (**c**) Hand-clapping (0, 12). (**d**) Hand-clapping (3, 15).

**Figure 7 biomimetics-09-00175-f007:**
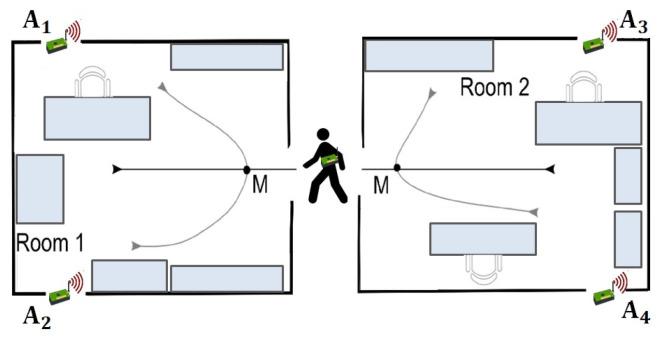
Simplified illustration of the type of user trajectories. There are six trajectory types, and M is the context point where the decision is made.

**Figure 8 biomimetics-09-00175-f008:**
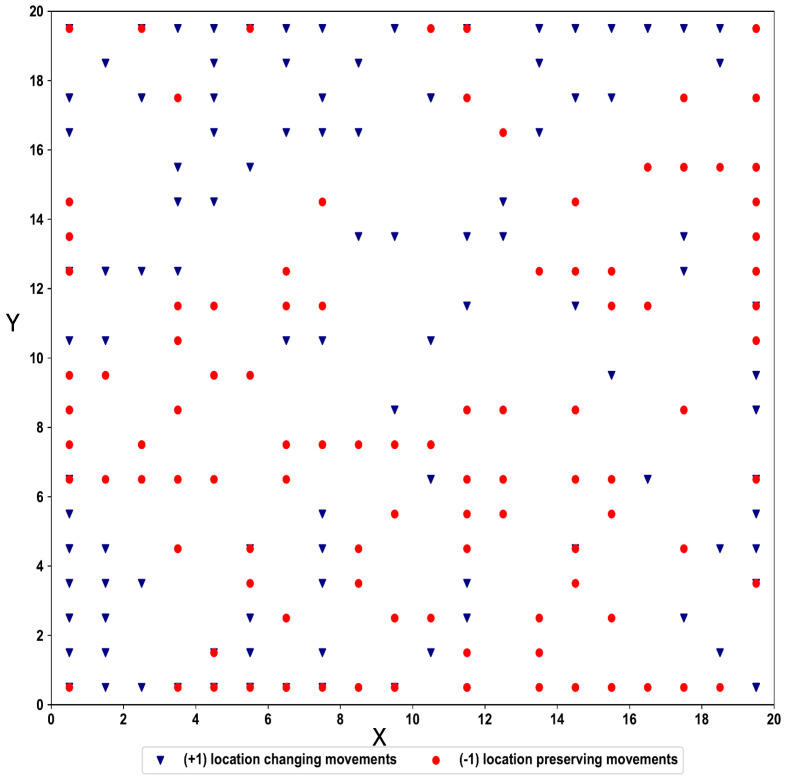
Indoor movement cluster output using Spatio-Temporal-SOM.

**Figure 9 biomimetics-09-00175-f009:**
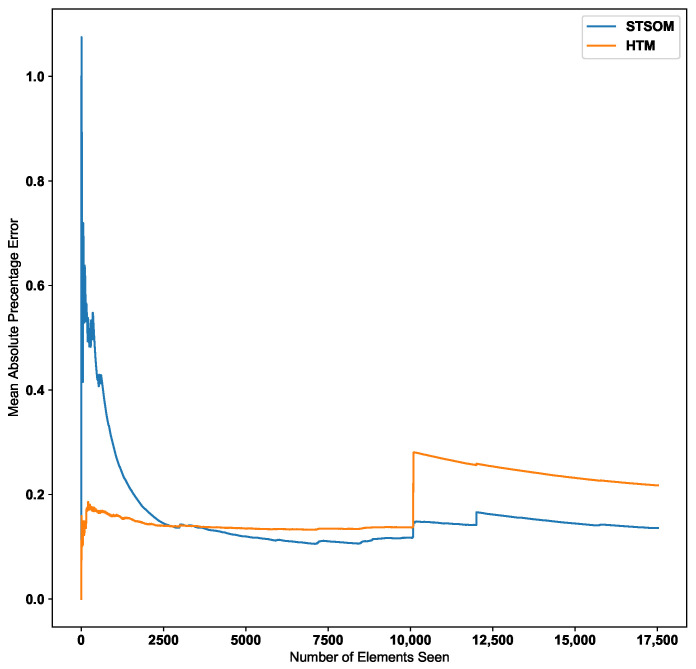
HTM algorithm vs. Spatio-Temporal-SOM error values for Taxi passenger dataset.

**Figure 10 biomimetics-09-00175-f010:**
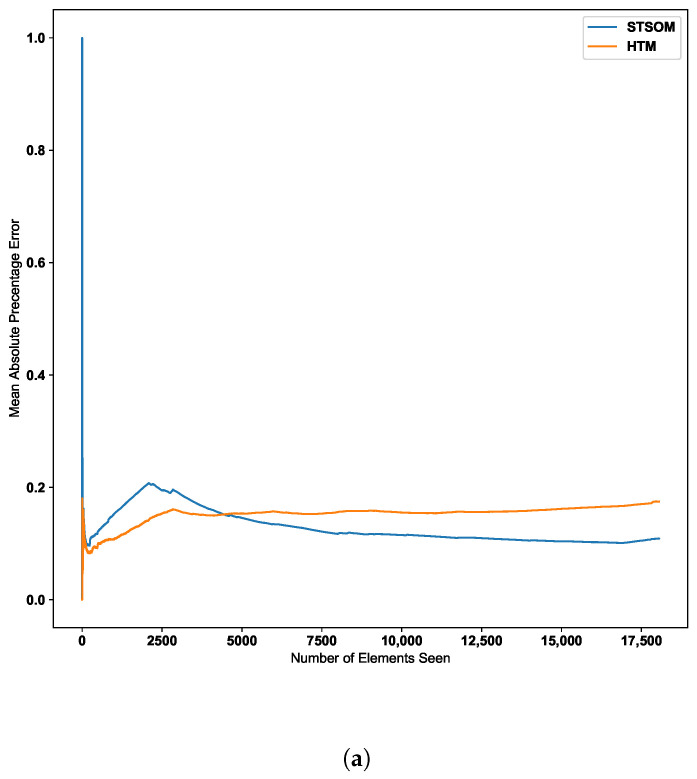
HTM algorithm vs. Spatio-Temporal-SOM error values for CPU usage and machine temperature datasets. (**a**) CPU usage. (**b**) Machine temperature.

**Table 1 biomimetics-09-00175-t001:** KTH accuracy comparison.

Proposed Approach	Method	Acc
Yang et al. (2013) [53]	Unsupervised	91.0
Peng et al. (2018) [54]	Unsupervised	83.4
Proposed ST-SOM	Unsupervised	86.6

**Table 2 biomimetics-09-00175-t002:** Selected KTH handclapping sequences.

Cluster Node	Selected Sequences
(0, 19)	542, 576, 542, 542
(0, 18)	542, 576, 576, 542
(0, 12)	542, 576, 542, 576, 576, 542
(3, 15)	668, 668, 542, 668, 668, 542

**Table 3 biomimetics-09-00175-t003:** Indoor movement accuracy comparison.

Approach	Method	Acc
D. Bacciu et al. (2011) [56]	Supervised	89.5
Ours	Unsupervised	87.3

**Table 4 biomimetics-09-00175-t004:** Indoor movement hyperparameters.

Hyperparameter	Value
First-layer learning rate	0.1
Second-layer learning rate	0.5
Number of iterations	100

**Table 5 biomimetics-09-00175-t005:** Aggregated MAPE error values.

Dataset	HTM	Spatio-Temporal-SOM
Taxi passenger	0.217	**0.119** ^1^
CPU usage	0.175	**0.109** ^1^
Machine temperature	0.0197	**0.0192** ^1^

^1^ **Bold** text indicate lowest MAPE values

## Data Availability

Data are contained within the article.

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
