# Peer review of "A Two-Layer Self-Organizing Map with Vector Symbolic Architecture for Spatiotemporal Sequence Learning and Prediction"

_biomimetics, 2024, doi:10.3390/biomimetics9030175_

Round 1

Reviewer 1 Report

Comments and Suggestions for Authors

Stream data analysis and processing sensory information is an important task in the field of data science and robotics. This paper proposed an interesting method for Spatiotemporal Sequence Learning and Prediction using Self Organizing Map. The authors stated that their proposed method uses a two-layer structure in which the first one accepts the individual elements from input sequences and the second one uses an encoded representation of the whole sequence using VSA considering the time dimension. An item memory links the sequence to generate a sequence of hyperdimensional vectors of fixed dimension. Despite the fascinating idea of this research, there are some issues that need to be clarified and revised to improve the article:

-         Representing a general flowchart or schematic diagram is very useful to describe the method as the authors provide it. Still, it is necessary to provide the pseudo-code for the learning and prediction phase of the method. The label prediction from memory is the only pseudo-code that is provided which is not enough.

- The authors didn’t explain the learning scheme of the network clearly. Are both layers trained simultaneously or not? The method applied to analyze spatio-temporal data stream and the learning scheme should be clearly (step-wise) described.

- It is stated that the proposed method is capable of representing fast-changing dynamic spatiotemporal patterns. Accordingly, the proposed SOM method needs to have the capability to adapt itself for possible changes in the data manifold that need variable neighborhood function and learning rate which are not considered in the proposed approach. The authors should clarify this issue.

-         It is declared that the topology-preserving neighborhood of the SOMs provides the ability to identify potential alternative spatial elements and temporal sequences, but how do the authors assure the quality of mapping? Measures like topographic function or topographic error could be considered to evaluate the topology preservation in both network layers.

-         How the size of the network in both layers (number of neurons) is decided?

-         The SOM network with a lower learning rate can also converge to the appropriate answer with a higher number of epochs. Thus, the weaker performance of the method at low learning rates can be due to an insufficient number of epochs. This issue should be clarified.

-         Most plots are not of good quality and the axis labels and tick-marks are illegible.

-      Variable “xi” in equation 1 was not introduced in the text.

According to the mentioned issues, I recommend a major revision of this article.

Comments on the Quality of English Language

-   "X" character should not be used instead of the multiplication sign.

Reviewer 2 Report

Comments and Suggestions for Authors

I have several suggestions and comments to the authors:

1. The introduction section is too big. I suggest shorter introduction concentrated on the major goal of the paper.

2. According to me the manuscript needs an example for the application of the suggested machine learning strategy to a real data set from monitoring data - the prediction in this case is of utmost importance.

3. The authors should address in more details the significance of hierarchical clustering as contribution to the suggested new architecture for spatiotemporal prediction ( a simple approach integrated in a complex algorithm with SOM)

Reviewer 3 Report

Comments and Suggestions for Authors

The paper proposes a new nature and neuro-science-inspired algorithm for spatiotemporal learning and prediction based on sequential recall and vector symbolic architecture, with three steps of algorithm state, however:

1. The author states in the abstract that "The algorithm has been empirically evaluated using benchmark and real-world datasets to demonstrate its advantages compared to the state-of-the-art in spatiotemporal sequence learning.", but It is not shown in the manuscript.

2. It is nice to have statistical results in the abstract.

3. Please add current references in the introduction to further clarify the background of what you want to achieve.

4. Why is there "Section 2.2 explain HTM algorithm in detail?" Inside the section 2.2.

5. Please add additional/current references in section "2. Related Work", with the initial origin of information and achieved result by the researcher.

6. Please add a Notation for each equation.

7. Missing X and Y axis information for Figures 5, 8.

8. "Conclusion" does not show whether the proposed method is acceptable or not.

9. Please add discussion and future works in the Conclusion section.

Comments on the Quality of English Language

Minor editing of English needed.

Round 2

Reviewer 1 Report

Comments and Suggestions for Authors

The authors have addressed all the comments and improved the manuscript's quality and clarity. Accordingly, I recommend the acceptance of the paper for the publication